# Error Analysis of Normal Surface Measurements Based on Multiple Laser Displacement Sensors

**DOI:** 10.3390/s24072059

**Published:** 2024-03-23

**Authors:** Fantong Meng, Guolin Yang, Jiankun Yang, Haibo Lu, Zhigang Dong, Renke Kang, Dongming Guo, Yan Qin

**Affiliations:** 1State Key Laboratory of High-Performance Precision Manufacturing, Dalian University of Technology, Dalian 116024, China; 825093006@mail.dlut.edu.cn (F.M.);; 2Pengcheng Laboratory, Shenzhen 518055, China

**Keywords:** normal measurement accuracy, laser displacement sensor, rotationally symmetric layout, random sampling, assembly hole

## Abstract

The robotic drilling of assembly holes is a crucial process in aerospace manufacturing, in which measuring the normal of the workpiece surface is a key step to guide the robot to the correct pose and guarantee the perpendicularity of the hole axis. Multiple laser displacement sensors can be used to satisfy the portable and in-site measurement requirements, but there is still a lack of accurate analysis and layout design. In this paper, a simplified parametric method is proposed for multi-sensor normal measurement devices with a symmetrical layout, using three parameters: the sensor number, the laser beam slant angle, and the laser spot distribution radius. A normal measurement error distribution simulation method considering the random sensor errors is proposed. The measurement error distribution laws at different sensor numbers, the laser beam slant angle, and the laser spot distribution radius are revealed as a pyramid-like region. The influential factors on normal measurement accuracy, such as sensor accuracy, quantity and installation position, are analyzed by a simulation and verified experimentally on a five-axis precision machine tool. The results show that increasing the laser beam slant angle and laser spot distribution radius significantly reduces the normal measurement errors. With the laser beam slant angle ≥15° and the laser spot distribution radius ≥19 mm, the normal measurement error falls below 0.05°, ensuring normal accuracy in robotic drilling.

## 1. Introduction

Bolting and riveting are widely used in aircraft assembly, with up to approximately two million assembly holes in a large aircraft [1]. Assembly quality significantly affects the fatigue life and stability of aircraft components because the assembly regions are weak links in aircraft structures and are prone to concentration stress. In total, 50–90% of the fatigue failures of aircraft components occur around assembly holes [2,3]. During the hole-making process, the normal error of the tool relative to the workpiece surface significantly affects assembly accuracy. Based on finite element analysis and fatigue tests, the fatigue life of aerospace components can be reduced by 7.7% and 14.9% when the normal error of the assembly hole reaches ±5° and ±10°, respectively [4,5]. The normal accuracy of the assembly hole is typically required as ±0.5° in the aerospace assembly process [6].

The assembly hole drilling of aircraft components commonly adopts in situ machining, which is usually operated by workers using pneumatic drilling tools [7,8]. With the characteristics of high flexibility and spatial reachability, automated drilling systems based on industrial robots are gradually being applied in aircraft assembly. To guarantee the normal accuracy of machining holes, different kinds of measuring devices are installed at the end of the drilling robots to detect the normal of the workpiece surface and guide the robot to the correct posture. Zhang et al. used four contact displacement sensors installed on the drilling end effector to achieve the normal vector of the workpiece surface [9]. Zhang and Sa et al. achieved the fusion 3D-vision measurement method for the normal accuracy of assembly holes and countersink on aircraft components using cameras and laser sensors, respectively [10,11]. Zhang and Bi et al. used two orthogonal linear laser sensors to measure the normal of the workpiece surface, thus adapting the normal adjustment requirements for robot drilling on complex curved surfaces [12].

Laser displacement sensors with high accuracy and small size that use a non-contact measurement method are widely used for the normal measurement of workpiece surfaces in the process of assembly hole drilling. Mei and Yuan et al. integrated four laser displacement sensors symmetrically arranged around the tool axis on a robotic drilling end effector for the normal measurement of aircraft components [13,14]. Lin et al. designed a flexible pressing foot for the drilling robot to adhere to the workpiece surface, and three laser sensors were used to measure the backplane of the pressing foot, thereby indirectly measuring the workpiece surface’s normal direction [15]. Gao and Tian et al. used four symmetrically arranged laser sensors to form a normal measurement device, making the laser beams parallel to the tool axis, which is called the rhombus layout [16,17].

The normal measurement error of the multi-laser-sensor measurement device is mainly caused by the sensor measurement error, sensor installation parameter error, and fitting error of the normal direction on the complex curved surface. Regarding the in situ measurement laser displacement sensor at the end of the machining system, Wang et al. proposed a modified chi-square and Gaussian process prediction method to analyze and separate the primary sources of the measurement error [18]. Chen and Zhang et al. used tools, such as the laser tracker and gauge blocks, to calibrate and compensate for the installation error of laser sensors [19,20]. Gao et al. analyzed the influence of the workpiece surface curvature on normal measurement accuracy when using a multi-laser-sensor device in the normal measurement of aircraft components [21]. The measurement error of the laser sensor directly affects the normal measurement accuracy. Sun and Zhang et al. analyzed the 3D vision measurement model of the laser displacement sensor and calibrated the measurement error of the sensor with a specialized calibration block with a complex 3D optical structure, respectively [22,23]. In the measurement error analysis of the laser displacement sensor, Nan et al. and Chen et al. constructed the Scheimpflug measurement model and achieved the sensor calibration using the mutation operator-based particle swarm optimization algorithm and the Monte Carlo optimization strategy, respectively [24,25].

In this study, a multi-sensor normal measurement device is modeled and parameterized based on its structural characteristics, and both the distribution and variation principle of the normal measurement error caused by the sensor measurement errors are analyzed. In the rest of the paper, a multi-sensor normal measurement device with a rotationally symmetric layout is proposed, and the normal measurement process is provided in Section 2; a normal measurement error distribution model based on the random sampling of sensor errors is proposed, and the influential principle of factors such as sensor accuracy, geometric parameters of the measured plane and the structure of the normal measurement device on the normal measurement accuracy are separately analyzed in Section 3; in Section 4, a normal measurement experimental platform based on a five-axis precision machine tool is set up, and the variety principle of the normal measurement accuracy is verified through experiments; the results and discussion are given in Section 5; and the conclusion is provided in Section 6.

## 2. Normal Measurement Principle of Rotationally Symmetric Multi-Sensor Measurements

### 2.1. Normal Measurement Based on Multiple Laser Sensors with Rotationally Symmetric Layout

Laser displacement sensors based on non-contact laser triangulation take advantage of high accuracy and reliability. The front surface of the sensor is known as the datum plane, where the origin point of the sensor is located, as shown in Figure 1. In the measurement process of the laser displacement sensor, the laser beam is illuminated through a fixed path from the origin point to the workpiece surface to form a laser spot. The reflected light passes through the sensor lens group and the images on the CMOS. The distance from the sensor’s origin point to the laser spot is the value of the laser sensor. The main technical parameters of the laser displacement sensor include measurement range and accuracy. The measurement range is defined by the maximum measurement distance *l*_max_ and minimum measurement distance *l*_min_, where the sensor range is denoted as *l*_r_ = *l*_max_ − *l*_min_, and the sensor mid-range value is *l*_mid_ = (*l*_max_ + *l*_min_)/2. The measurement accuracy of the laser sensor is affected by repeatability *e*_r_ and linearity *e*_l_. The measurement error of the laser sensor is denoted as *l*_e_ = *e*_r_ + *e*_l_ × *l*_r_.

A normal measurement device composed of multiple laser sensors is generally integrated at the end of a drilling robot or a machine tool. In the normal measurement process, while multiple laser sensors illuminate the workpiece surface, the position of the laser spots is calculated according to the sensor installation parameters and measured values, and the normal of the workpiece surface is calculated using the position of multiple laser spots. A geometric layout of multiple laser sensors with rotational symmetry is proposed for the modeling and parameterization of the multi-sensor measurement device, as shown in Figure 2. The measurement coordinate system fixed to the measurement device is defined with the origin point of each laser sensor located on the X-Y plane and the laser beams symmetrically rotated around the Z-axis. The sensor quantity of the multi-sensor normal measurement device is denoted as *n*, and the angle between the laser beam and the Z-axis is called the laser beam slant angle, which is denoted as *γ*. A plane perpendicular to the Z-axis of the measurement coordinate system is defined as the reference plane of the normal measurement device, and the measurement value of each sensor to this plane is *l*_mid_. Laser spots on the reference plane are located on the same circle, which is called the laser spot distribution circle, and the radius of the laser spot distribution circle is denoted as *r*. These three parameters, *n*, *γ*, *r*, define the geometric structure of the multi-sensor normal measurement device. Technical parameters and the corresponding expression symbols of the normal measurement device are listed in Table 1.

### 2.2. Normal Calculation Based on Parameter-Matrix Eigen-Decomposition

The measurement model of each laser sensor in the multi-sensor normal measurement device can be abstracted as a space ray model, and the laser sensor installation parameters can be expressed as follows:(1)$$S_{i}=[x_{i},\;y_{i},\;z_{i},\;\alpha_{i},\;\beta_{i}{]}^{T}\;\in \;R^{5},i=1,2,\ldots ,n$$where [*x_i_*, *y_i_*, *z_i_*] is the origin point of sensor *i*, *α_i_* is the angle between the X-axis of the measurement coordinate system and the projection of the laser beam on the X-Y plane, and *β_i_* is the angle between the laser beam and the Z-axis.

With the rotationally symmetrical geometric layout of the normal measurement device, each parameter of the laser sensors can be derived as follows: the origin point of each sensor is defined in the X-Y plane of the measurement coordinate system, whereas the angle between the laser beam and the Z-axis is defined as *γ*. Therefore, *z_i_* = 0, *β_i_* = *γ*, *i* = 1, 2, …, *n*. The installation parameters of all laser sensors are shown in Equation (2).
(2)$$S_{i}=\left[\begin{array}{c} \left(r+l_{i}\cos(\gamma ))\cdot \cos(2\pi (i-1)/n\right)\\ \left(r+l_{i}\cos(\gamma ))\cdot \sin(2\pi (i-1)/n\right)\\ 0\\ \begin{array}{c} 2\pi (i-1)/n\\ \gamma \end{array} \end{array}\right],i=1,2,\cdots ,n$$

The measurement value of each laser sensor is denoted as *l_i_*, *i* = 1, 2, …, *n*. Using the parameters and values of a laser sensor, the position of laser spots on the measured plane is calculated as ***p****_I_* = [*p_ix_*, *p_iy_*, *p_iz_*], *i* = 1, 2, …, *n*, which is expressed explicitly as follows:(3)$$p_{i}=\left[\begin{array}{c} \left(r+l_{i}\cos(\gamma ))\cdot \cos(2\pi (i-1)/n)+l_{i}\cdot \cos(2\pi (i-1)/n)\cdot \sin(\gamma \right)\\ \left(r+l_{i}\cos(\gamma ))\cdot \sin(2\pi (i-1)/n)+l_{i}\cdot s\mathrm{in}(2\pi (i-1)/n)\cdot \sin(\gamma \right)\\ l_{i}\cdot \cos(\gamma ) \end{array}\right]$$

The positions of the laser spots are used to establish equations for the normal vector of the measured plane. While a unique solution can be obtained when *n* = 3, the least squares method is used to solve the overdetermined equations when n > 3. Using the least square method to calculate the normal vector via laser spots, the average value of the spots is first calculated, which is denoted as $\bar{p}=\left[{\bar{p}}_{x},{\bar{p}}_{y},{\bar{p}}_{z}\right]=1/n\cdot [\sum_{i=1}^{n}p_{ix},\sum_{i=1}^{n}p_{iy},\sum_{i=1}^{n}p_{iz}]$. This leads to the setup of matrix ***A***,
(4)$$A=\left(\begin{array}{c} P^{1}-\bar{P}\\ P^{2}-\bar{P}\\ \vdots \\ P^{n}-\bar{P} \end{array}\right)=\left(\begin{array}{ccc} p_{x}^{1}-{\bar{p}}_{x} & p_{y}^{1}-{\bar{p}}_{y} & p_{z}^{1}-{\bar{p}}_{z}\\ p_{x}^{2}-{\bar{p}}_{x} & p_{y}^{2}-{\bar{p}}_{y} & p_{z}^{2}-{\bar{p}}_{z}\\ & \vdots & \\ p_{x}^{n}-{\bar{p}}_{x} & p_{y}^{n}-{\bar{p}}_{y} & p_{z}^{n}-{\bar{p}}_{z} \end{array}\right)$$

The normal vector ***v*** is calculated by solving the equation *v* = argmin***_v_*** (||***A*** · ***v***||_2_). The optimization problem is equivalent to the least squares solution of the equation,
***A*** · ***v*** = 0(5)
which is achieved based on the eigen-decomposition method. The matrix ***M*** = ***A***^T^ · ***A*** is set as follows:(6)$$M=\left(\begin{array}{ccc} \sum_{i=1}^{n}{\left(p_{x}^{i}-{\bar{p}}_{x}\right)}^{2} & \sum_{i=1}^{n}\left(p_{x}^{i}-{\bar{p}}_{x}\right)\left(p_{y}^{i}-{\bar{p}}_{y}\right) & \sum_{i=1}^{n}\left(p_{x}^{i}-{\bar{p}}_{x}\right)\left(p_{z}^{i}-{\bar{p}}_{z}\right)\\ \sum_{i=1}^{n}\left(p_{x}^{i}-{\bar{p}}_{x}\right)\left(p_{y}^{i}-{\bar{p}}_{y}\right) & \sum_{i=1}^{n}{\left(p_{y}^{i}-{\bar{p}}_{y}\right)}^{2} & \sum_{i=1}^{n}\left(p_{y}^{i}-{\bar{p}}_{y}\right)\left(p_{z}^{i}-{\bar{p}}_{z}\right)\\ \sum_{i=1}^{n}\left(p_{x}^{i}-{\bar{p}}_{x}\right)\left(p_{z}^{i}-{\bar{p}}_{z}\right) & \sum_{i=1}^{n}\left(p_{y}^{i}-{\bar{p}}_{y}\right)\left(p_{z}^{i}-{\bar{p}}_{z}\right) & \sum_{i=1}^{n}{\left(p_{z}^{i}-{\bar{p}}_{z}\right)}^{2} \end{array}\right)$$

Matrix ***M*** is a symmetric matrix with eigen-decomposition as ***M*** = ***Q**Λ**Q***^T^, where ***Q*** is an orthogonal matrix and ***Λ*** is a three-dimensional vector. Let ***Λ*** take a descending structure, denoted as ***Λ*** = [*λ*_1_, *λ*_2_, *λ*_3_]^T^, |*λ*_1_| ≥ |*λ*_2_|≥ |*λ*_3_| > 0. According to Equation (5), ***M*** = ***A***^T^ · ***A***, ***M*** · ***v*** = 0. The equation is then as follows,
***QΛQ***^T^ · ***v*** = 0(7)

With set ***y*** = ***Q***^T^ · ***v***, Equation (7) can be expressed as ***QΛy*** = 0. While ***Q*** ≠ 0,
***Λ*** · ***y*** = [*λ*_1_, *λ*_2_, *λ*_3_]^T^ · ***v*** = 0(8)

According to the descending structure of Λ, the least squares solution to Equation (8) is given as
***y*** = [0, 0, *y*_3_]^T^ = argmin***_y_*** (***Λ*** · ***y***)(9)
and the normal vector ***v*** can be calculated as
(10)$$v=Q\cdot y=\left[\begin{array}{ccc} Q_{11} & Q_{12} & Q_{13}\\ Q_{21} & Q_{22} & Q_{23}\\ Q_{31} & Q_{32} & Q_{33} \end{array}\right]\cdot \left[\begin{array}{c} 0\\ 0\\ y_{3} \end{array}\right]$$
(11)$$v=[Q_{13},\;Q_{23},Q_{33}{]}^{T}\;\cdot \;y_{3}$$

The unit normal vector *v* can be given as
***v*** = [*Q*_13_, *Q*_23_, *Q*_33_]^T^/||[*Q*_13_, *Q*_23_, *Q*_33_]||_2_(12)

### 2.3. Error Propagation of Sensor Measurements for Normal Calculation

The measurement value of laser sensors participates in the normal calculation process directly, and the existence of any sensor measurement error inevitably leads to the fitting error of the normal vector. As shown in Equation (3), the sensor error *l_i_*_e_ initially causes the coordinate error of the laser spot ***P****_i_*_e_.
(13)$$p_{ie}=\left[\begin{array}{c} \left(l_{i}+l_{ie})\cdot \cos(2\pi (i-1)/n)\cdot (\cos(\gamma )+\sin(\gamma ))+r\cos(2\pi (i-1)/n\right)\\ \left(l_{i}+l_{ie})\cdot s\mathrm{in}(2\pi (i-1)/n)\cdot (\cos(\gamma )+\sin(\gamma ))+rs\mathrm{in}(2\pi (i-1)/n\right)\\ \left(l_{i}+l_{ie})\cdot \cos(\gamma \right) \end{array}\right]$$

The laser spots’ coordinate error is then propagated into matrix ***A***, as in Equation (4). The normal vector is obtained by solving the system of homogeneous equations presented in Equation (5). Thus, errors in coefficient matrix ***A*** led to a decrease in the solution accuracy of the equation set and resulted in errors in the normal vector calculation.

As presented in Equation (13), the sensor error affects all components of the laser spot coordinate, which has a direct impact the normal measurement accuracy; the structural parameters *n*, *γ*, *r* simultaneously impact on the *p_ix_* and *p_iy_* components of the laser spot coordinates, whereas *p_iz_* is solely influenced by *γ*. All structural parameters of the normal measurement device have the potential to affect the error propagation from the sensor measurement to normal vector estimation. However, due to the non-linear process of the normal calculation, as demonstrated in Equations (6)–(12), the theoretical analysis of the effect of either the sensor’s accuracy or the device’s structural parameters on the normal measurement error via numerical methods is rather difficult.

## 3. Normal Measurement Error Estimation

The measurement error of laser sensors inevitably causes normal measurement errors. A normal measurement error estimation method based on random sampling is proposed to analyze the influence of factors such as sensor accuracy and the structure of multi-sensor measurement devices on normal measurement accuracy.

### 3.1. Normal Measurement Error Estimation Method Based on Random Sampling

Laser displacement sensors that are commonly used in multi-sensor normal measurement devices are usually off-the-shelf products. Under the specified permissible measurement environment, the measurement accuracy of the sensors is calibrated and guaranteed by the manufacturer. Sensor measurement error *l*_e_, as shown in Table 1, consists of linearity and repeatability errors, indicating the maximum limitation of the sensor error that may occur. In practical measurement processes, the sensor error can be any value within the error range [−*l*_e_, *l*_e_], according to the sensor technical manual. 

According to the normal vector calculation method shown in Equations (3)–(12), the normal vector of the measured surface is obtained based on the sensor measurement values. Utilizing this computational process, an imitation of the real measurement process can be achieved by injecting artificial errors into the sensors, and the normal vector with the error is calculated. In this section, a method for analyzing the normal measurement accuracy through random sampling is proposed. First, the structural parameters n, γ, and *r* of the multi-sensor measurement device are defined, and the spatial equation of the measured plane, a*x* + b*y* + c*z* + d = 0, is determined. As shown in Equation (3), the coordinates of the laser spot on the measured surface can be derived based on the parameters *n*, *γ*, *r*, and the sensor measurement value *l_i_*. The laser spot coordinates, at the same time, fulfill the equation of the measured plane, which subsequently leads to the calculation of the sensor theoretical measurement value *l_i_*^Theo^:(14)$$l_{i}^{Theo}=-\frac{arcos(2\pi (i-1/n))+brsin(2\pi (i-1/n))+d}{\left(acos(2\pi (i-1/n))+bsin(2\pi (i-1/n)))(cos(\gamma )+sin(\gamma ))+ccos(\gamma \right)},i=1,2\ldots ,n$$

This is shown in Figure 3a. After obtaining the theoretical measurement value for sensors, a random measurement error within the sensor error range is selected for each sensor. The laser spots calculated with sensor values that include measurement errors also carry coordinate errors, which lead to a discrepancy between the fitted normal vector and the theoretical normal of the measured plane, resulting in normal measurement errors, as depicted in Figure 3b. Repeating the process of the random sampling of sensor measurement errors and recomputing the normal vectors, a group of varying normal measurement results with different errors are gradually obtained, as illustrated in Figure 3c. Furthermore, as the continuous sampling of diverse instances within the sensor error range is conducted to compute normal vectors, the accumulated normal measurement results progressively approximate the actual distribution of measurement errors, and the maximum normal measurement error *n*_emax_ indicates normal measurement accuracy. The complete process of the normal measurement error estimation method based on random sampling is illustrated in Figure 4.

To validate the analysis method for the distribution interval of normal measurement errors, a simulation process of normal measurement error analysis based on random sampling is implemented. In this analytical case, a normal measurement device based on four laser sensors was employed, in which the laser beam slant angle was 0°, and the laser spot distribution radius was 20 mm. Each laser sensor in the measurement device has a measurement range of 100 mm ± 30 mm, with a linear accuracy of 0.15% and a repeatability error of 0.005 mm. Considering the premise that insufficient iterations can affect the estimation accuracy of normal measurement errors, multiple simulation tests were conducted on the number of iterations in this case. The simulation results revealed that, upon reaching an approximate cycle of 50,000 iterations, the normal measurement error *n*_emax_ showed no significant further improvement. Consequently, the number of iterations *N* was set to 50,000 in the following simulation analysis.

The distribution of normal vectors and their two-dimensional projection are shown in Figure 5a. The normal error exhibits a pyramid-like special distribution. The maximum error *n*_emax_ equals 0.152° and represents the normal measurement accuracy, as shown in Figure 5b. The normal vector is typically expressed as a unit vector; nonetheless, for the clear visualization of inherent patterns in normal measurement errors within its distribution region and its corresponding 2D projection, a length of 1 mm was adopted for the calculated normal vector. Consequently, in Figure 5a,b, the dimension of length is annotated accordingly. The extent of normal measurement errors in later simulation studies is readily conveyed by the 2D projection of the normal measurement error distribution region.

Normal measurement errors caused by the sensor errors can be estimated using the simulation method based on the random sampling of sensor errors. Furthermore, the proposed method is used to analyze the influence patterns of factors such as the pose of the measured plane, laser sensor accuracy, and geometric parameters of the measurement device on the normal measurement accuracy.

### 3.2. Normal Measurement Accuracy Influenced by the Measurement Plane Position

Using a multi-sensor normal measurement device to obtain the normal of the workpiece surface, the position and posture of the measured plane are limited within a fixed area due to the range limitation of the laser sensors. In the measurement coordinate system, the position of the measured plane is constrained by the distance between the plane and the coordinate origin, which is denoted as the plane distance, and the plane direction is constrained by the angle between the plane normal and the coordinate Z-axis which is denoted as a normal slant, as shown in Figure 6.

A normal measurement device is constructed using four laser displacement sensors arranged in a rotationally symmetric layout, with the laser beam slant angle *γ* set to 0° and the laser spot distribution radius *r* set to 15 mm. The measurement range of the laser sensor is set to 100 mm ± 30 mm, with a linear accuracy of 0.15% and a repeatability error of 0.005 mm. By ensuring that the theoretical measurement value of each sensor is within this range and by varying the plane distance and normal slant of the measured plane, the influential patterns of sensor errors on the normal measurement accuracy are analyzed using the simulation method based on random sampling. The result is shown in Figure 7. During the variation in the plane distance and normal slant, the normal measurement error caused by sensor errors consistently remains within the range of [0.15°, 0.17°]. In the aerospace assembly process, the normal error of the assembly hole is mainly required to be less than 0.5°, and the current measurement error, which reaches a 1/3 of the machining precision, significantly affects the assembly holes’ quality.

### 3.3. Normal Measurement Error Influenced by Sensor Accuracy

For a multi-sensor measurement device, the normal measurement error caused by sensor errors is not affected by the position or attitude of the measured plane. The reference plane of the normal measurement device is set as the measured plane when the influence of other factors on the normal measurement accuracy is investigated. The measurement error of the laser displacement sensor, including the linearity error and repeatability error, has the most direct impact on the normal measurement accuracy. A normal measurement device is constructed using four laser displacement sensors arranged in a rotationally symmetric layout, where the laser beam slant angle *γ* is set to 0°, the laser spot distribution radius *r* is set to 15 mm, and the measurement range of the laser sensor set to 100 mm ± 30 mm. The sensor linearity error is calculated based on the linear accuracy and the measurement range given by Equation (3), selecting the linearity error within the region of [0.01 mm, 0.065 mm] and the repeatability error within the range of [0.001 mm, 0.01 mm]; the influence patterns of sensor measurement errors on the normal measurement accuracy are analyzed by the proposed random sampling simulation method, which is shown in Figure 8.

According to the simulation results, enhancing the linearity or repeatability can both contribute to a decrease in the normal measurement error, and the influence of these factors on the normal measurement accuracy follows a linear relationship. When the linearity error is less than 0.02 mm, the repeatable error is less than 0.006 mm, and the normal measurement error caused by sensor errors is controlled within 0.15°, as shown in Figure 8, which meets the precision requirement of normal measurement in the aerospace assembly process. However, the pursuit of sensor precision has resulted in a significant increase in cost and size, making it difficult to reconcile with the automatic drilling of assembly holes.

### 3.4. Normal Error Influenced by Normal Measurement Device

The laser beams of the multi-sensor normal measurement device illuminate the workpiece surface, forming multiple laser spots. Using the coordinates of these laser spots, a plane is fitted using the least squares method to obtain the normal vector of the workpiece surface. In the measurement process, sensor errors lead to incorrect calculations of the laser spot coordinates, which results in a normal measurement error. Sensor installation parameters which are involved in the calculation process of the laser spot coordinates, as shown in Equations (1)–(3), affect the normal measurement error as well. In the multi-sensor normal measurement device with a rotationally symmetric layout, the position and attitude of each sensor are determined by the number of sensors *n*, the laser beam slant angle *γ*, and the laser spot distribution radius *r*. In this section, the influential patterns of these three parameters on the normal measurement accuracy are analyzed.

#### 3.4.1. Normal Errors Influenced by Laser Sensor Quantity

The multi-sensor normal measurement device with a rotationally symmetric layout has sensors evenly distributed around the Z-axis of the measurement coordinate system, and sensor quantity directly affects the position of each sensor. The measurement range of the laser displacement sensor is set to 100 mm ± 30 mm, with a linearity of 0.15% f.s. and a repeatability error of 0.005 mm. With the laser beam slant angle of the measurement device set to 0° and the laser spot distribution radius set to 15 mm, the influence pattern of the sensor quantity on the normal measurement accuracy can be analyzed, as shown in Figure 9.

With an increase in the sensor quantity, the normal measurement accuracy gradually improves, as shown in Figure 9. Furthermore, the spatial distribution of the normal measurement error is obtained based on the simulation analysis method with the random sampling of sensor errors. As shown in Figure 10, when using three sensors, the cross-section of the normal error distribution approximates a hexagon; when using four sensors, the normal error distribution has a square-like cross-section; and as the sensor quantity increases continuously, the spatial distribution of the normal error is similar to that of a circle. Although normal vectors are inherently dimensionless, a length of 1 mm is assigned to each normal vector, as visualized in Figure 10. Based on this representation, it becomes visually apparent that a smaller-size normal vector distribution region corresponds to a higher normal measurement accuracy. 

#### 3.4.2. Normal Error Influenced by Laser Beam Slant

The direction of the laser displacement sensor is determined by the laser beam slant angle, which is a key parameter in calculating the laser spot coordinate, as shown in Equation (3). Four laser sensors with a measurement range of 100 mm ± 30 mm were selected to constitute a normal measurement device, with a linearity of 0.15% and a repeatability of 0.005 mm. The laser spot distribution radius *r* of the measurement device was set to 15 mm, and the laser beam slant angle *γ* increased from 0° to 45°, as shown in Figure 11. The proposed simulation method based on random sampling was used to obtain the varying pattern of the normal measurement error.

The influence pattern of the laser beam slant angle on the normal measurement error is shown in Figure 12. When *γ* gradually increases from 0°, the normal measurement accuracy improves significantly; after *γ* reaches 16°, the normal measurement accuracy reaches 0.05° with the improvement in accuracy continually slowing down.

#### 3.4.3. Normal Error Influenced by Laser Spot Distribution

Laser spots, which are located on the reference plane of a normal measurement device, are arranged in a circular configuration. The laser spot distribution radius, which reflects the positioning of the laser sensors, is another crucial parameter employed in the calculation of the laser spot coordinates, as shown in Equation (3). Four laser displacement sensors with a measurement range of 100 mm ± 30 mm were used in the multi-sensor normal measurement device, with a linearity of 0.15% and repeatability of 0.005 mm. With the laser beam slant angle set to 0°, the laser spot distribution radius increased from 2 mm to 25 mm, as shown in Figure 13, and the simulation method based on sensor error random sampling was used to analyze the variation pattern of the normal measurement accuracy.

The effect of the laser spot distribution radius on the normal measurement error is shown in Figure 14. As the laser spot distribution radius increased gradually from 2 mm, the normal measurement accuracy experienced a significant enhancement, albeit with a concomitant gradual decline in the rate of improvement.

#### 3.4.4. Normal Error Coupling Influenced by Both Laser Beam Slant and Laser Spot Distribution

Geometric parameters of the multi-sensor normal measurement device participate in the calculation of laser spot coordinates, significantly influencing the normal measurement errors induced by sensor errors. An increase in the number of sensors, the laser beam slant angle and the laser spot distribution radius improve the accuracy of the normal measurement. However, in the automatic machining process of assembly holes in aerospace components, the normal measurement device is generally mounted at the end of the machining spindle on a drilling robot or machine tool, and the size constraints make it highly challenging to increase the sensors’ quantity. According to the proposed rotationally symmetric structure, driving a laser sensor to rotate around the Z axis of the measurement coordinate system and measuring the workpiece surface at fixed angular intervals aligns with the principle of the multi-sensor normal measurement device, which also enables the mitigation of the effects of sensor variabilities on the normal measurement accuracy. However, the number of assembly holes is usually massive in the manufacturing of aerospace components, and a multi-sensor normal measurement device is essential to guarantee drilling efficiency in the aerospace assembly process. Therefore, further research into methods that ensure the high accuracy of normal measurements remains necessary with only a few laser sensors. In this section, the coupling effect of both the laser beam slant angle *γ* and the laser spot distribution radius *r* on the normal measurement error is analyzed with the simulation method based on sensor errors random sampling. 

The coupling influential patterns of the laser beam slant angle *γ* and the laser spot distribution radius *r* on normal measurement accuracy are shown in Figure 15. When *γ* reaches 5°, and *r* reaches 7 mm, the normal measurement error is less than 0.15°, which satisfies the normal measurement requirements in the assembly hole machining process of aerospace components; when *γ* reaches 15°, and *r* reaches 19 mm, the normal measurement error is less than 0.05°, which has little impact on the normal accuracy of assembly holes. 

## 4. Experimental Verification

### 4.1. Experimental Platform

In an actual normal measurement process, the normal accuracy was affected by multiple factors, such as the installation error of the sensors or the fitting error of the curved measured surface. To reveal the impact of sensor errors on normal measurement accuracy and avoid interference from other factors, a special normal measurement experimental platform was necessary.

According to the rotationally symmetric structure of the multi-sensor normal measurement system, the laser beams and origin points of the sensors are symmetrically rotated around the Z-axis of the measurement coordinate system. In the experiment, the scheme of a laser sensor rotating around a fixed central axis was used instead of the multi-sensor framework to build the experimental platform, as shown in Figure 16. Using this method, the consistency of sensor installation parameters in the measurement device was ensured, thereby avoiding the normal measurement errors arising from installation errors and disparities in sensor precision. An OMRON ZX2-L100 laser displacement sensor (OMRON Corporation, Kyoto, Japan) with a measurement range of 100 mm ± 35 mm was used to build the experimental platform. The repeatability accuracy of the sensor is 0.005 mm, and the linearity accuracy is 0.15% of the full scale. The sufficient measurement range and accuracy make it representative of the normal measurement process. To simulate the conditions of different laser beam directions and laser spot distribution radiuses of the multi-sensor normal measurement device, it was necessary to accurately adjust the position and attitude of the sensor. A pose adjustment platform with 2DOF was used for this purpose, which consisted of a goniometer stage and a linear stage. The sensor was installed on the platform using a sensor holder.

To ensure the rotational accuracy of the measurement module’s central axis and to provide a precisely perpendicular measured plane to that axis, a DMG ULTRASONIC 50 precision five-axis machine tool was used in the experiment (DMG Mori AG, Bielefeld, Germany). The positioning accuracy of the machine tool was 0.005 mm, which ensured that the sensors on the measurement platform were aligned accurately with the test plane. The 2-DOF pose adjustment platform was installed on the machine tool rotation platform through a permanent magnetic chuck (Longbo Co., Ltd., Shanghai, China). A special HSK-63A toolholder (BTSK, Jining, China) was customized with a 60 mm diameter cylinder at the head, and the end face of the cylinder had vertical accuracy with the axis of the toolholder. We installed the toolholder on the machine tool spindle, and the end face was used as the measured surface, which is accurately perpendicular to the central axis, and the height is also known. The complete experimental platform is shown in Figure 17.

Before the normal measurement test, it was necessary to accurately adjust the position and posture of the point laser sensor based on the theoretical values of the laser beam slant angle and laser spot distribution radius of the multi-sensor measurement modules used in the experiment. As shown in Figure 18, during the installation and adjustment of the 2-DOF platform, the contact probe equipped on the machine tool was used to measure the installation surfaces on the platform and the sensor holder directly, ensuring that the sensor installation was in accordance with the theoretical pose.

### 4.2. Experimental Process

A multi-sensor normal measurement device composed of a single laser displacement sensor rotating around the central axis was used in the normal measuring experiment, exploring the influential law of the laser beam slant angle and the laser spot distribution radius on the normal measurement accuracy. The measured plane was the front face of a customized toolholder with a cylindrical head, which was perpendicular to the central axis of the normal measurement device. The sensor quantity of the normal measurement device was set to 4; that is, the distance was measured with the laser displacement sensor when the machine tool turntable was at the specific positions of 0°, 90°, 180°, and 270°, respectively.

In the experiment that verified the influence of the laser beam slant angle on the normal measurement accuracy, three groups of tests were performed with the laser beam slant angles *γ* of 0°, 5°, and 10° alongside a laser spot distribution radius of 8 mm. While *γ* was adjusted precisely to the goniometer stage, a change in *r* was caused as well. Therefore, the linear stage was also required to compensate *r* to 8 mm. In each group of experiments, the normal of the measured plane was repeatedly measured 10 times. The normal direction of the plane was fitted using the sensor measurement values, and the normal measurement error was calculated as the angle between this fitted normal and the actual normal direction. According to the experimental platform shown in Figure 15, the actual normal of the measured plane *n*_p_ was parallel to the Z axis of the measurement coordinate system. See Table 2.

In the experiment on the influence of the laser spot distribution radius on normal measurement accuracy, the goniometer stage was repeatedly adjusted while the machine tool probe was used to ensure that the laser beam slant angle was 0°. Then, the linear stage was used to change the laser spot distribution radius *r*. Three groups of tests were conducted when *r* equaled 4 mm, 8 mm, and 12 mm, and the normal direction was repeatedly measured 10 times in each group of the test. The normal error is the angle between the measured normal and the actual normal of the measured plane. See Table 3.

While the normal measurement errors induced by sensor errors are random, the maximum normal error of each group of experiments is selected to analyze the influential patterns of the laser beam slant angle and laser spot distribution radius on the normal measurement accuracy, as shown in Figure 19. Furthermore, the experimental results are then compared with the simulation results based on the sensor errors in random sampling. According to the experimental results, the increase in both the laser beam slant angle and the laser spot distribution radius leads to a reduction in normal measurement errors, and this trend is consistent with the simulation results. However, the normal measurement errors from the experimental results are smaller than those from the simulation results. This is attributed to the limited number of experiments and is insufficient to obtain the complete distribution of the normal measurement error.

## 5. Result and Discussion

In the normal measurement process of the multi-sensor measurement device, the normal measurement error caused by the laser sensor error is random; however, the normal error distribution is obtained. The rotationally symmetric structure of the proposed normal measurement device is constrained by the following three parameters: the number of sensors *n*, the laser beam slant angle *γ*, and the laser spot distribution radius *r*. A method for analyzing the normal measurement error of a multi-sensor measurement device based on the random sampling of sensor errors is proposed. The normal measurement error is distributed in a pyramid-like spatial region, and the maximum error represents the measurement accuracy. 

According to the simulation results, changes in the position and attitude of the measured plane have little effect on the normal measurement error, while improvements in laser sensor accuracy directly improve the normal measurement accuracy. The error propagation of the sensor measurement in normal calculation was analyzed in Section 2.3. The sensor accuracy directly impacts the error range of the laser spot coordinates, hence exerting a significant influence on the normal measurement error; meanwhile, the position and orientation of the measured plane have little effect on the laser spot precision. These reasons elucidate the simulation analysis outcomes presented above. However, even high-precision laser sensors with a linearity accuracy of up to 0.03 mm and repeatability accuracy of 0.001 mm may still cause a normal measurement error greater than 0.15°, which significantly affects the normal accuracy of assembly holes. 

The geometric parameters of a normal measurement device can also affect the normal measurement accuracy for all three petameters—*n*, *γ* and *r*—and participate in the computation of the laser spot coordinates. Increasing the number of sensors can gradually improve the normal measurement accuracy while also changing the shape of the normal measurement error distribution. Due to the involvement of the sensor number *n* in the laser spot coordinates calculation, sensor measurement errors can lead to the rotation of the laser spots around the Z-axis of the measurement coordinate system, while the error associated with the normal vector fitting is not directly impacted. However, in the normal fitting process using the least squares method, changes in *n* directly affect the number of samples being fitted, which consequently leads to the results displayed in the simulation data. According to the simulation analysis results, increasing either the laser beam slant angle or the laser spot distribution radius can significantly improve the normal measurement accuracy. In the laser spot calculation, the spot distribution radius *r* is independent of the sensor values *l_i_*. However, an increase in *r* indeed results in a separation between the laser spots, which consequently reduces the sensitivity of the normal vector fitting process to errors in the spot coordinates. On the contrary, there exists a significant relationship between the laser beam slant angle *γ* and the sensor values *l_i_* in the laser spot calculation. As the optical path deviation angle increases, the impact of measurement errors on the errors in spot coordinates decreases. The above reasons are considered to give rise to the results exhibited in the simulation data.

The influence of these two factors on the normal measurement accuracy is further verified on the experimental platform based on a five-axis precision machine tool and OMRON ZX2-L100 laser displacement sensor. Using the proposed simulation method to analyze the coupling effect of the laser beam slant angle and the laser spot distribution radius on the measurement accuracy of a four-sensor normal measurement device, with the laser beam slant angle ≥ 5° and the laser spot distribution radius ≥ 7 mm, the normal measurement error is less than 0.15°, which can meet the requirement of assembly hole drilling; with the laser beam slant angle ≥ 15° and the laser spot distribution radius ≥ 19 mm, the normal measurement error is less than 0.05°, which has negligible effect on normal measurements.

## 6. Conclusions

This study analyzed the normal measurement errors induced by sensor errors in multi-sensor measurement devices and proposed a simulation method based on sensor errors random sampling to analyze the influencing factors. After thorough simulation analyses and experimental validations, the following conclusions are drawn:(1)Laser sensor precision effect: the simulation results demonstrate that an increase in laser sensor precision enhances the normal measurement accuracy.(2)Structural parameters’ significance: The three geometric parameters of the normal measurement device, including the number of sensors *n*, the laser beam slant angle *γ*, and laser spot distribution radius *r*, significantly influence the normal measurement accuracy. The increase in parameters *n*, *γ* and *r* all contribute to enhancing the normal measurement accuracy.(3)Layout optimization of the normal measurement device: For a four-sensor normal measurement device, when the laser beam slant angle is ≥5°, the laser spot distribution radius is ≥7 mm, and the normal measurement error can be maintained within 0.15°, thereby satisfying the requirements for drilling assembly holes. When the laser beam slant angle is ≥15°, the laser spot distribution radius is ≥19 mm, and the normal measurement error is reduced to ≤0.05°, resulting in a negligible effect on normal measurements.

Valuable guidance is provided for optimizing multi-sensor normal measurement devices to enhance their normal measurement accuracy and meet the precision requirements of assembly hole drilling in aerospace components.

## Figures and Tables

**Figure 1 sensors-24-02059-f001:**
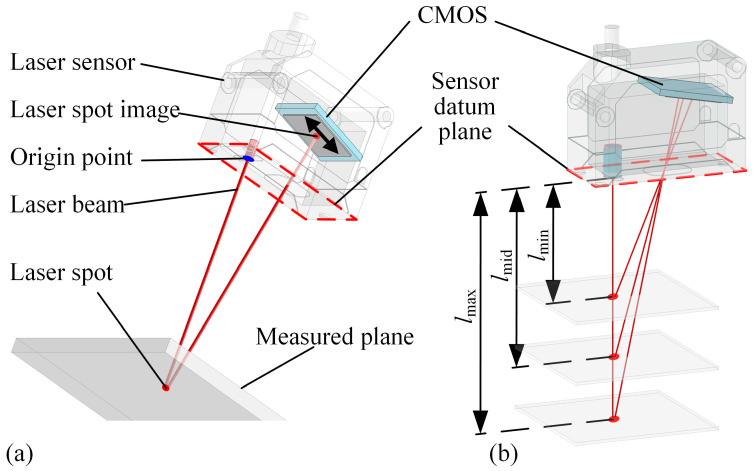
Measurement principle of laser sensor. (**a**) Distance measurement based on laser triangulation principle, (**b**) where the measurement value is the distance between the origin point and the laser spot. Between *l*_max_ and *l*_min_, the measurement range of the sensor can be identified, and *l*_mid_ is the median measurement value.

**Figure 2 sensors-24-02059-f002:**
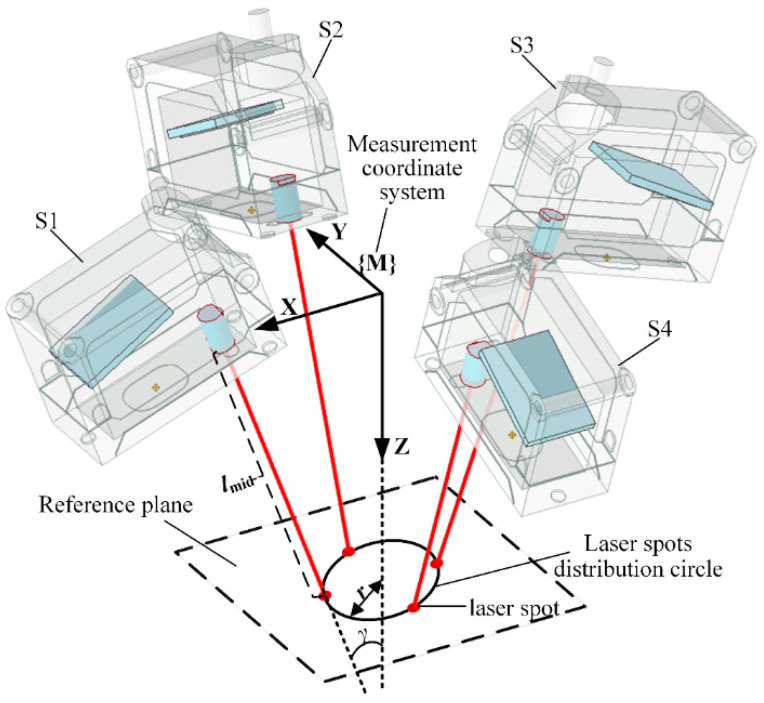
Rotationally symmetric multi-sensor normal measurement system with laser beam symmetry around the Z-axis of the measurement coordinate system.

**Figure 3 sensors-24-02059-f003:**
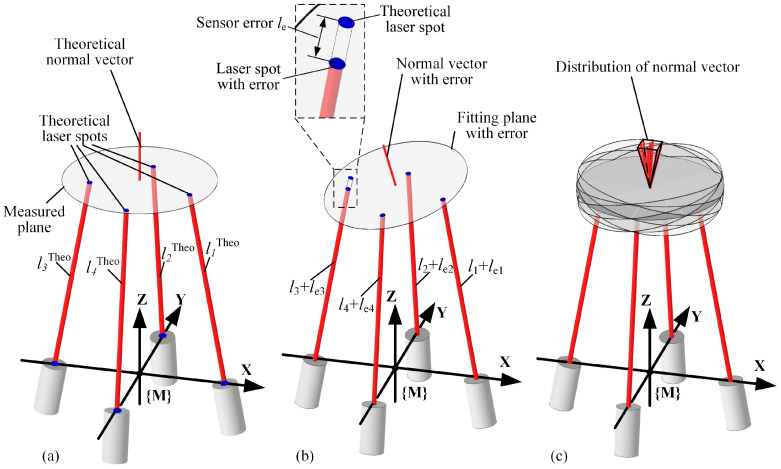
Generation principle of normal errors caused by sensor errors. (**a**) Theoretical normal vector fitted by the multi-sensor measurement system without sensor errors. (**b**) Normal fitting error caused by sensor errors. (**c**) The randomness of sensor errors leads to the dispersion of normal errors in space.

**Figure 4 sensors-24-02059-f004:**
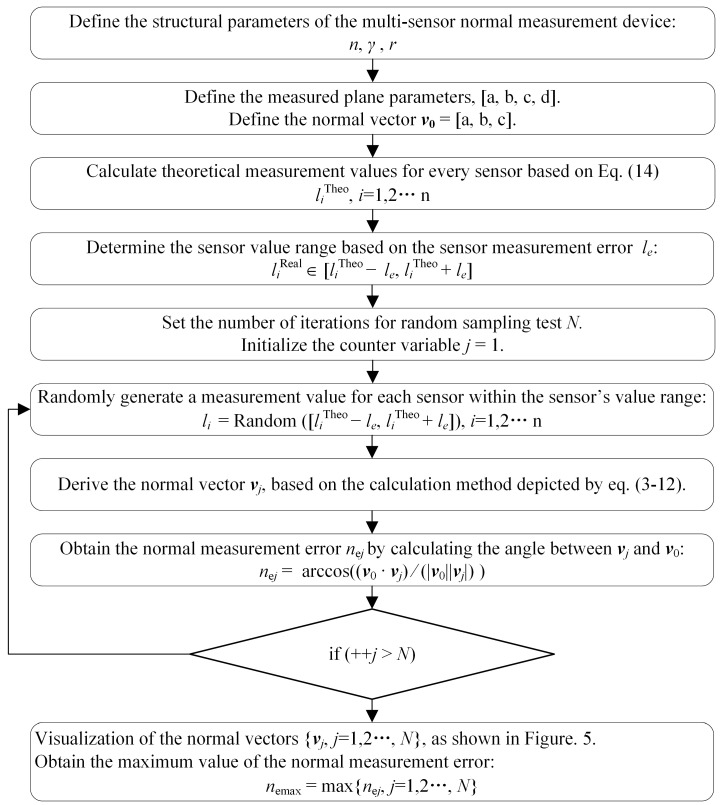
Flowchart of normal measurement error estimation method based on random sampling.

**Figure 5 sensors-24-02059-f005:**
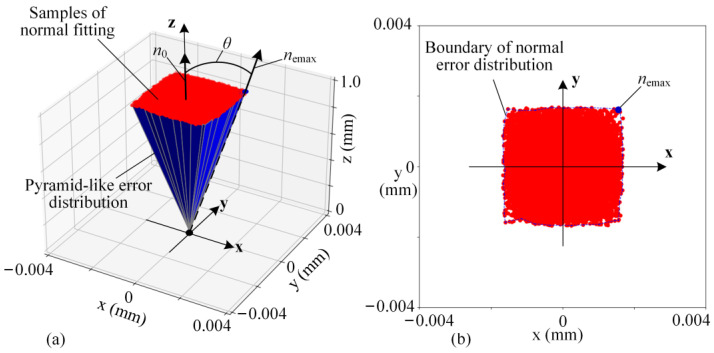
Normal error distribution based on the random sampling method. (**a**) In the measurement coordinate system of the multi-sensor normal measurement system, the normal error caused by sensor errors follows a pyramid-like distribution. (**b**) The maximum error found in the XOY plane represents the normal measurement accuracy, denoted as *n*_emax_.

**Figure 6 sensors-24-02059-f006:**
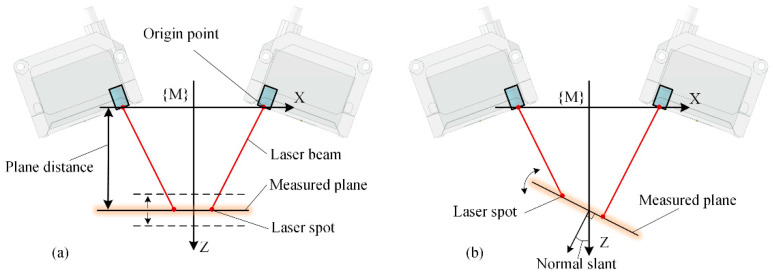
Normal measurement process with the changes in the measured planes. (**a**) The distance between the measured plane and the XOY plane of the measurement coordinate system is the plane distance. (**b**) The angle between the normal of the measured plane and the Z-axis of the measurement coordinate system is the normal slant.

**Figure 7 sensors-24-02059-f007:**
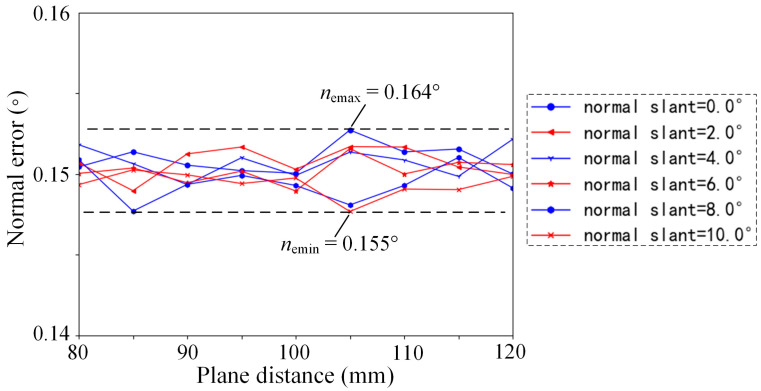
Normal measurement error influenced by the plane distance and normal slant of the measured planes.

**Figure 8 sensors-24-02059-f008:**
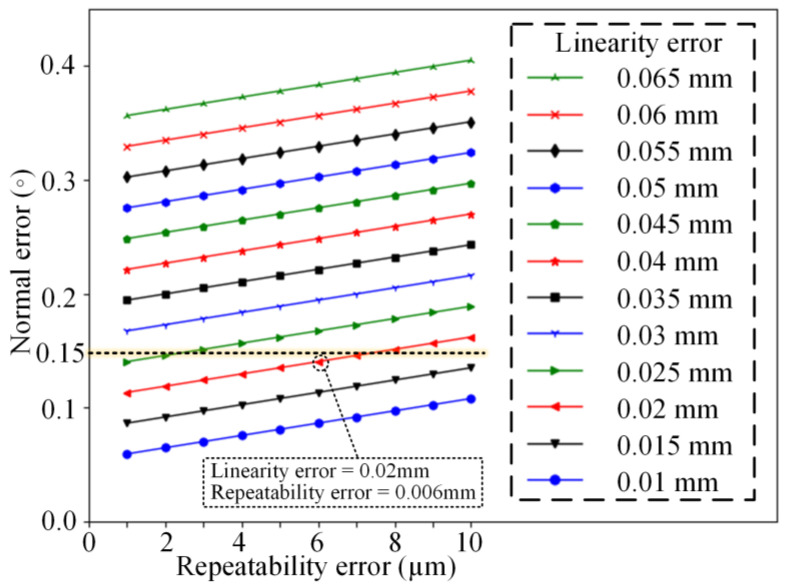
Normal measurement error influenced by sensor errors.

**Figure 9 sensors-24-02059-f009:**
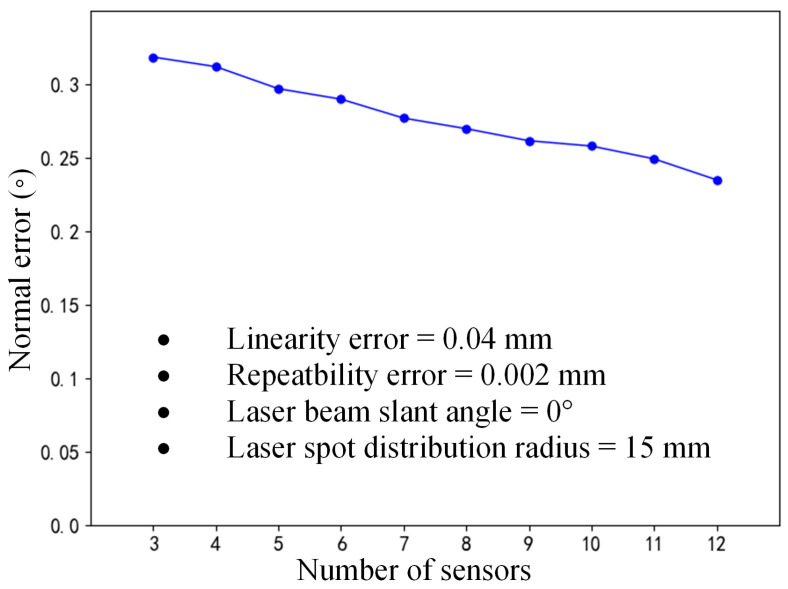
Normal measurement error influenced by the number of sensors.

**Figure 10 sensors-24-02059-f010:**
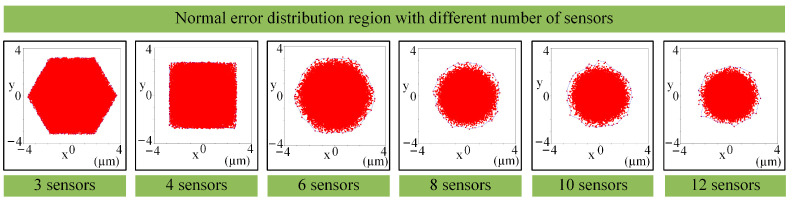
Normal error distribution of multi-sensor systems for different numbers of sensors.

**Figure 11 sensors-24-02059-f011:**
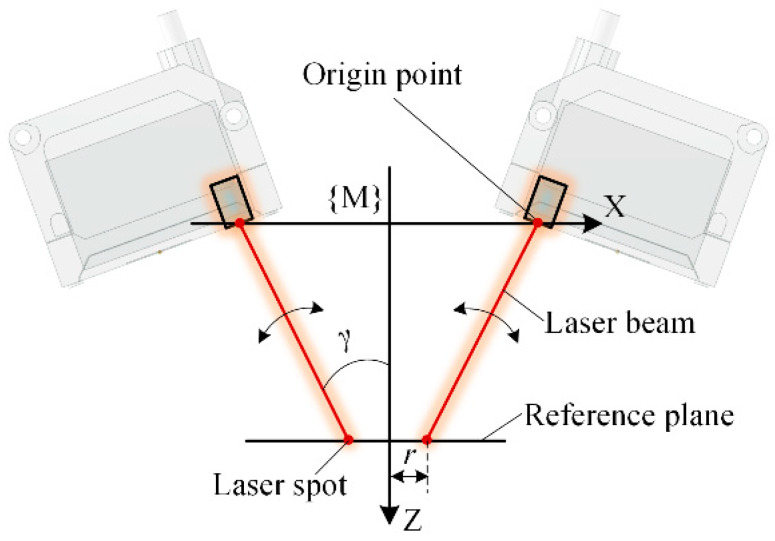
Normal measurement process with the changes in laser beam slant angle *γ*.

**Figure 12 sensors-24-02059-f012:**
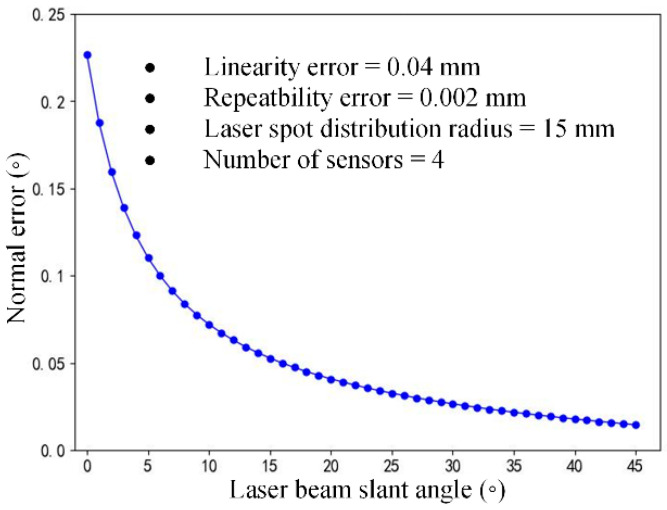
Normal measurement error influenced by the laser beam slant angle.

**Figure 13 sensors-24-02059-f013:**
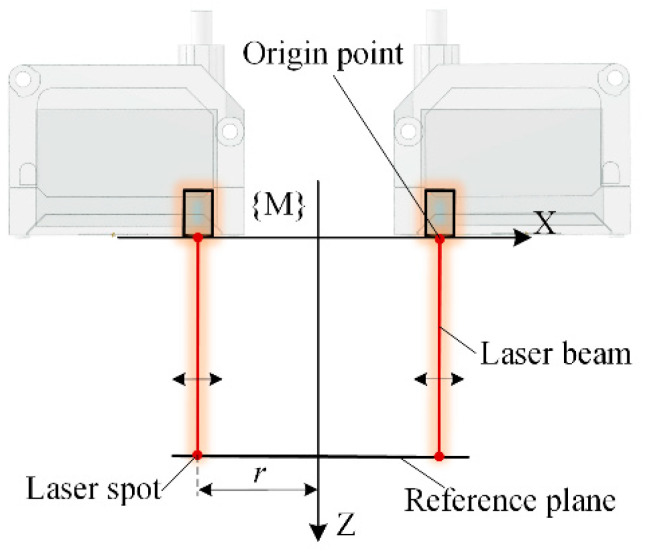
Normal measurement process with the changes in the laser spot distribution radius *r*.

**Figure 14 sensors-24-02059-f014:**
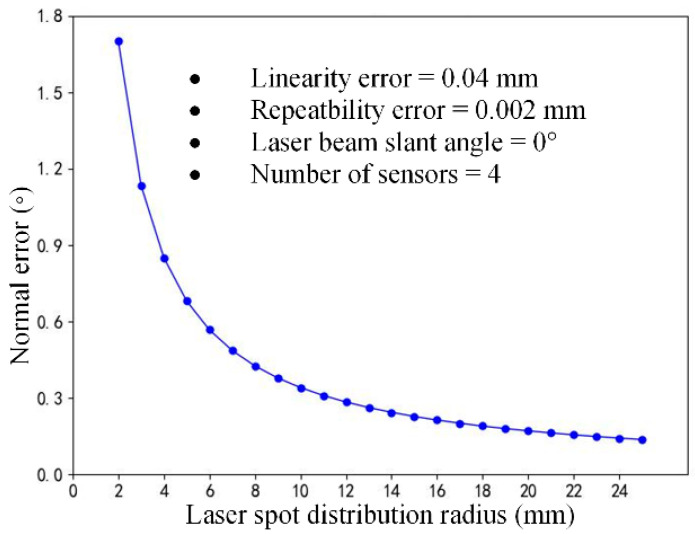
Normal measurement accuracy influenced by the laser spot distribution radius.

**Figure 15 sensors-24-02059-f015:**
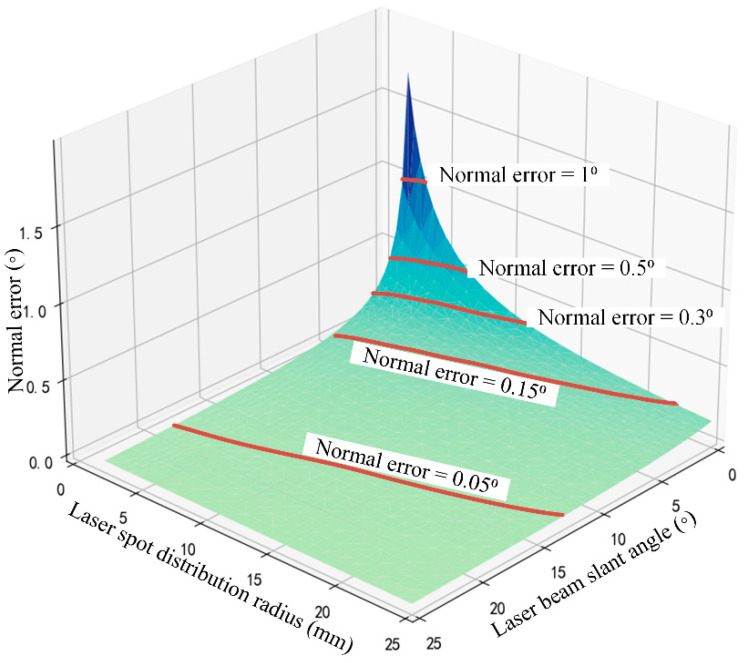
Normal measurement accuracy influenced by both laser beam slant angle and laser spot distribution radius.

**Figure 16 sensors-24-02059-f016:**
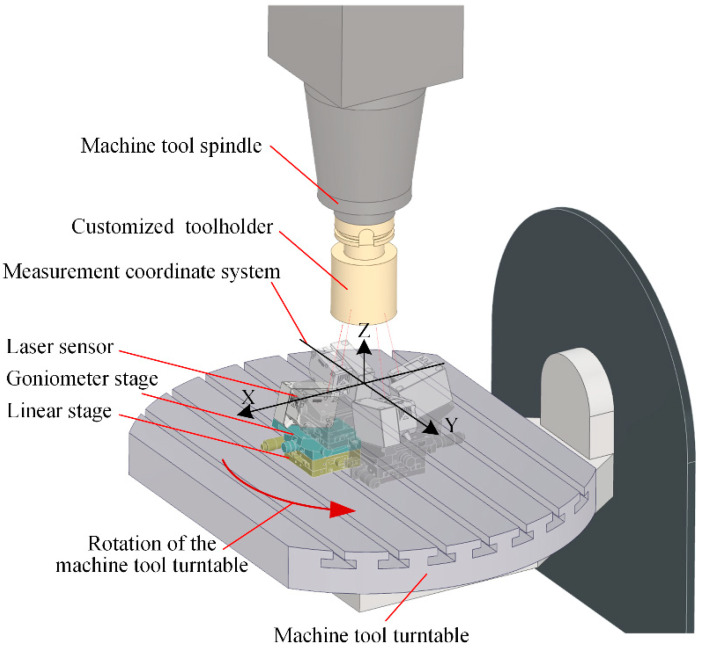
Normal measurement experiment with single laser sensor.

**Figure 17 sensors-24-02059-f017:**
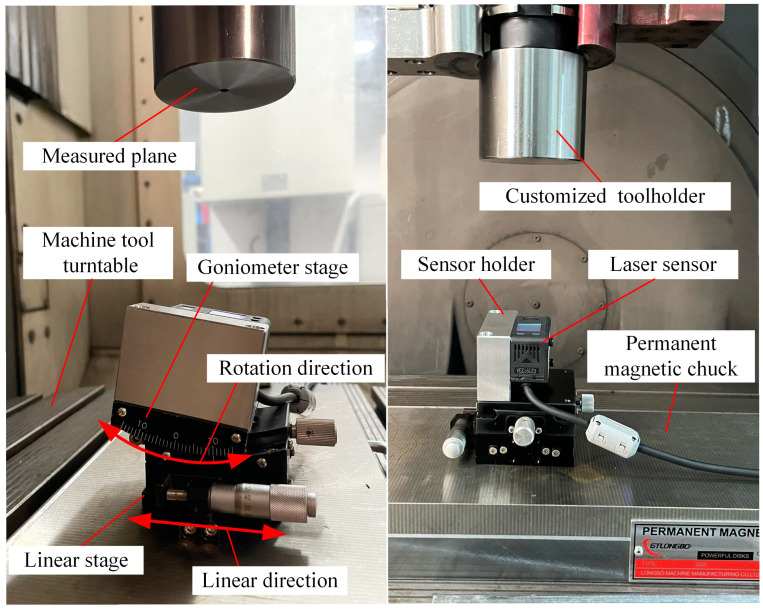
Normal measurement experimental platform based on the precision machine tool.

**Figure 18 sensors-24-02059-f018:**
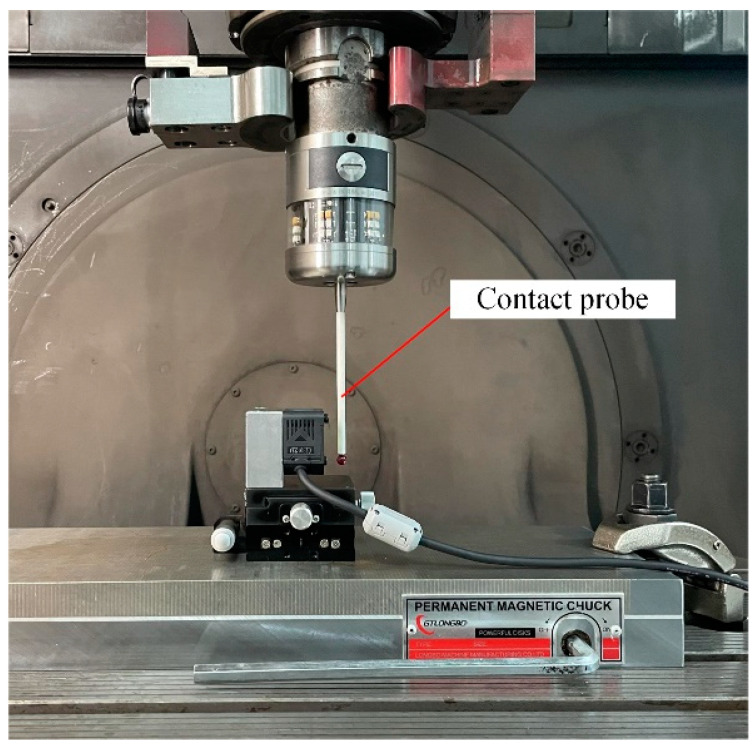
High precision probe used to assure the accurate pose of the laser sensor relative to the machine tool.

**Figure 19 sensors-24-02059-f019:**
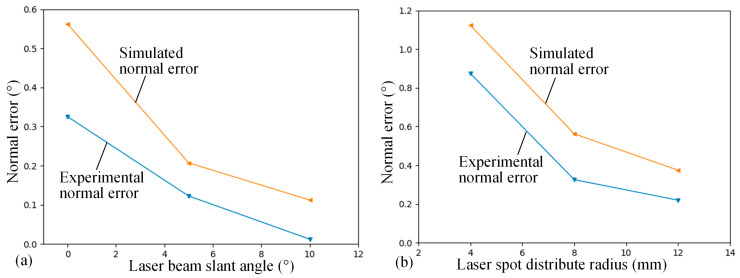
Experiment and simulation results of the normal measurement accuracy. (**a**) Normal measurement accuracy influenced by the laser beam slant angle. (**b**) Normal measurement accuracy influenced by the laser spot distribution radius.

**Table 1 sensors-24-02059-t001:** Parameters to describe the normal measurement device.

Technical Parameters	Expression Symbols
Maximum measurement distance of laser sensor	*l* _max_
Minimum measurement distance of laser sensor	*l* _min_
Median value of laser sensor measuring range	*l* _mid_
Measurement range of laser sensor	*l* _r_
Measurement error of laser sensor	*e*
Linearity of laser sensor	*e* _l_
Repeatability of laser sensor	*e* _r_
Number of laser sensors	*n*
The angle between the laser beam and the Z-axis of the measurement coordinate system	*γ*
Distribution radius of laser spot when each of the sensor values equals *l*_mid_	*r*

**Table 2 sensors-24-02059-t002:** Experimental results of different laser beam slant angles on the normal measurement accuracy.

No.	Slant Angle γ (°)	Experimental Normal Error (°)	Simulated Normal Error (°)
1	0	0.325	0.562
2	5	0.124	0.207
3	10	0.012	0.112

**Table 3 sensors-24-02059-t003:** Experimental results of different laser spot distribution radiuses on the normal measurement accuracy.

No.	Laser Spot Distribution Radius *r* (mm)	Experimental Normal Error (°)	Simulated Normal Error (°)
1	4	0.874	1.123
2	8	0.325	0.562
3	12	0.219	0.374

## Data Availability

Dataset available on request from the authors.

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
