# Peer review of "Error Analysis of Normal Surface Measurements Based on Multiple Laser Displacement Sensors"

_sensors, 2024, doi:10.3390/s24072059_

Round 1

Reviewer 1 Report

Comments and Suggestions for Authors

In this paper, normal measurement error induced by sensor errors in a multi-sensor measurement devices consisted by multiple laser displacement sensors is discussed by both simulation and experiment. The manuscript is reasonably organized. However, several issues need to be addressed before the manuscript can be recommended for publication.

1. The authors mentioned in the abstract that, a novel rotationally symmetric layout of multiple sensors .... However, symmetric layout of multiple sensors has traditionally used in displacement sensing such as laser triangulation measuring.

2. Whether the normal error distribution given in Fig. 4 is obtained by simulation or experiment? The authors should clarify in the text. The method of the simulation or experiment should by explained. How the data is obtained? And the parameters in the simulation or the experiment should also been given.

3. The theoretical analyze about the errors which is mentioned in Chap.3 is highly recommended to be offered in Chap.2. That is, the influence of these factors to the performance of the multi-sensors devices, such as accuracy or linearity, should be analyzed theoretically.

4. The comparison between the theoretical analyze and the simulated results should be given. For example, whether the results shown in Fig.6-8 are in agreement in the analyze in Chap.2? Figures obtained from equations mentioned in Chap.2 are suggested to be added to give a comparison.

5. In Fig.18, it shows a much smaller measured error compared to simulated error, which is hardly convinced to be explained by the insufficient experiment samples since this phenomenon happening at all slant angels. More experiment data is recommended to be offered to confirm the issue.

6. Several writing format errors should be corrected. For example, in line 119 in Figure 2 A ..., in line 295 sensor. The measurement..., in line 297 With the Laser ... and so on.

Comments on the Quality of English Language

English language is fine. Only few format errors should be corrected.

Reviewer 2 Report

Comments and Suggestions for Authors

The manuscript “Error analysis of surface normal measurement based on multiple laser displacement sensors” by Fantong Meng et al. presents results of the modeling and experimental study of the normal determination errors using assemblies of laser displacement sensors. The authors have considered the model for calculation of the surface normal based on the outputs of laser displacement sensors and provided simulations of measurement errors depending by various factors such as the number of sensors, sensor accuracy, uncertainty of the measurement plane position, etc. The obtained simulation results have been compared to the measurement data obtained using the designed experimental platform.  In general, this work is not a breakthrough research, but its results are of a certain interest from the point of view of the further development of sensor systems for robotics. However, the form of presentation and interpretation of the obtained results requires significant refinement before making a final decision on the acceptance for publication. The questions and comments are listed below.

1. “However, in practical operating of a laser sensor, its measurement error exhibit randomness within its nominal error range, which is unpredictable” (lines 187-188).

Generally speaking, this statement is incorrect; any set of erroneous data of a finite volume is characterized by a certain sample probability distribution. By introducing such generally accepted statistical parameters as the significance level or the confidence band, we can always determine the interval of error values in which the measured values will fall with a given probability determined by these parameters. It seems that in their analysis of errors the authors ignore the main criteria and estimates generally accepted in mathematical and applied statistics.

2. “To validate the analysis method for the distribution interval of normal measurement errors, in a measurement process of a normal measurement device based on 4 laser sensors, 50000 times of sensor errors random sampling and corresponding normal calculation are performed.” (lines 197-199).

This point requires significantly more detail in the description. Why was the number of statistical runs chosen to be 50000 rather than, for example, 40000 or 80000? How the errors were set for each of the 4 sensors; whether they were independent or correlated with each other? What type of random variables was used as input to the simulation - uniformly distributed, normally distributed, or some other? Justification for using this type is required. In general, a significantly more detailed description and justification of the modeling procedure is required.

3. In general, the images of the “normal error distribution regions” in Figures 4, b and 9 seem unrealistic (except, perhaps, for cases with the number of sensors 6 or more in Fig. 9, when the density of imaging points at the edges is less than in the center). Figures 4, b and 9 (the cases of 4 and 6 sensors) display the homogeneous distributions of the imaging points across the “normal error distribution regions”. As a rule, in real measuring systems, small random deviations are characterized by higher probabilities compared to large deviations; accordingly, one can expect that the density of imaging points will monotonically decrease from the centers of the regions to the edges. Uniform distributions of imaging points seem confusing. I recommend that the authors will use a different form of presentation of these areas; obviously, when displaying all 50,000 points in a limited area, they simply overlap each other and merge.

4. Figure 6 looks very bad; I recommend to move the legends to the figure caption and to display the area of interest with the sufficiently smaller scale along the vertical axis (e.g., from 0.14 degrees to 0.18 degrees). In addition, please, interpret the remarkable systematic error in measurements (displacement of the mean value with respect to the zeroth value).

5. Figure 18. The claimed reason for much larger values of simulated errors compared to the experimental data (“This is attributed to the limited number of experiments and is insufficient to obtain the complete distribution of normal measurement error.”, lines 467-468) seems inconsistent. On the one hand, who prevented the authors from conducting more experimental measurements? On the other hand, mathematical statistics show that when estimating the average values of random variables over a small data group, the mean value quickly converges to the expected magnitude as the number of data in the group increases. In my opinion, so great difference between the model and experimental data cannot be explained by “the limited number of experiments”.

6. In general, the English writing of the manuscript is appropriately clear and readable. Nevertheless, I recommend some additional editing (use of definite and indefinite articles, clarifying inserts (for instance, the title of subsection “3.2. Normal measurement accuracy influenced by the measurement plane”; it will be better “……by the measurement plane position”), etc.).

Comments on the Quality of English Language

Moderate editing of English language is required.

Reviewer 3 Report

Comments and Suggestions for Authors

Dear Professor

Concerns the MS entitled:  "Error analysis of surface normal measurement based on multiple laser displacement sensors" submitted for publication, I go through it and show that it interested and presented a new results on this topic.

There are some comments must release before considering as follows:

-Abstract must but be put in details to show the results obtained and includes a comparison between the previous and present results.

-Equations 1-4 must be cited.

-Figure 3 needs some details to show the differences between them.

-Figure 6, the y-axis "normal error" must be taken [0.14,0.18] to show its behavior.

-Figure 8 does not clear, authors must clear it.

-Physical meaning of figure 9 must be clear.

-Discussion and physical meaning of the results obtained must be put in details.

-Conclusion must be put in items to display the new results obtained.

Comments on the Quality of English Language

English must be revise to agree with the journal standard by professional in English.

Round 2

Reviewer 1 Report

Comments and Suggestions for Authors

The comments and questions have been solved resonably. I think the manuscript has been improved sufficiently

Reviewer 2 Report

Comments and Suggestions for Authors

The authors have provided a great bulk of work to improve the manuscript quality and scientific soundness. In its current form, the manuscript is acceptable for publication.